# Serious bacterial infections and antibiotic prescribing in primary care: cohort study using electronic health records in the UK

Martin C Gulliford [1,2] Xiaohui Sun [1] Judith Charlton,[1] Joanne R Winter,[1] Catey Bunce [1,2] Olga Boiko,[1] Robin Fox,[3] Paul Little,[4] Michael Moore,[4] Alastair D Hay,[5] Mark Ashworth [1] And SafeAB Research Group

[1]School of Population Health and Environmental Sciences, King's College London, London, UK
[2]NIHR Biomedical Research Centre at Guy's and St Thomas' Hospitals London, GreatMaze Pond, London, UK
[3]The Health Centre, Coker Close, Bicester, UK
[4]Primary Care Research Group, University of Southampton, Southampton, UK
[5]Centre for Academic Primary Care, Bristol Medical School, Population Health Sciences, University of Bristol, Bristol, UK

**Correspondence to**
Professor Martin C Gulliford;
martin.gulliford@kcl.ac.uk

## ABSTRACT

**Objective** This study evaluated whether serious bacterial infections are more frequent at family practices with lower antibiotic prescribing rates.

**Design** Cohort study.

**Setting** 706 UK family practices in the Clinical Practice Research Datalink from 2002 to 2017.

**Participants** 10.1 million registered patients with 69.3 million patient-years' follow-up.

**Exposures** All antibiotic prescriptions, subgroups of acute and repeat antibiotic prescriptions, and proportion of antibiotic prescriptions associated with specific-coded indications.

**Main outcome measures** First episodes of serious bacterial infections. Poisson models were fitted adjusting for age group, gender, comorbidity, deprivation, region and calendar year, with random intercepts representing family practice-specific estimates.

**Results** The age-standardised antibiotic prescribing rate per 1000 patient-years increased from 2002 (male 423; female 621) to 2012 (male 530; female 842) before declining to 2017 (male 449; female 753). The median family practice had an antibiotic prescribing rate of 648 per 1000 patient-years with 95% range for different practices of 430–1038 antibiotic prescriptions per 1000 patient-years. Specific coded indications were recorded for 58% of antibiotic prescriptions at the median family practice, the 95% range at different family practices was from 10% to 75%. There were 139 759 first episodes of serious bacterial infection. After adjusting for covariates and the proportion of coded consultations, there was no evidence that serious bacterial infections were lower at family practices with higher total antibiotic prescribing. The adjusted rate ratio for 20% higher total antibiotic prescribing was 1.03, (95% CI 1.00 to 1.06, p=0.074).

**Conclusions** We did not find population-level evidence that family practices with lower total antibiotic prescribing might have more frequent occurrence of serious bacterial infections overall. Improving the recording of infection episodes has potential to inform better antimicrobial stewardship in primary care.

## Strengths and limitations of this study

► This cohort study included 10.1 million patients with 69.3 million patient-years of follow-up at 706 UK family practices from 2002 to 2017.

► The study included all antibiotic prescriptions and classified them according to the medical conditions recorded on the same date.

► The study relied on medical conditions recorded by healthcare professionals in primary care.

► Missing and misclassified information might result in bias, which might generally be towards a null finding.

► The study aimed to evaluate associations at the general practice level and the results do not exclude the possibility of association at the individual patient level.

## INTRODUCTION

Antimicrobial resistance is a growing concern for health systems. The G20 health ministers noted that 'drug-resistant (organisms) are to blame for 700 000 deaths worldwide each year, and this figure is predicted to rise to 10 million by 2050 if urgent action is not taken.'[1] There are now intense efforts to reduce unnecessary use of antibiotics, especially in primary care where 80% of antibiotics are prescribed. These antimicrobial stewardship programmes have met with some success. In England, the total quantity of antibiotics prescribed in primary care declined by 13.2% in the 5 years between 2013 and 2017.[2 3] Bacterial infections are still of public health importance with 1.7 million cases of sepsis and 270 000 deaths per year in the USA.[4] Strategies to reduce inappropriate use of antibiotics must ensure that antibiotics can be used when they are needed.[5 6]

It is possible that reducing antibiotic prescribing might be associated with greater

risk of serious bacterial infections. Previous research investigated infection risk and antibiotic prescribing for respiratory illnesses.[3 7] In a cohort study, Petersen *et al*[8] found that antibiotic treatment reduced risks of mastoiditis after otitis media, peritonsillar abscess after sore throat and pneumonia after respiratory infection. An analysis of electronic health records[9] found that family practices that prescribed antibiotic more frequently to patients with self-limiting respiratory illnesses might have lower risk of pneumonia and peritonsillar abscess but there were no associations with risk of mastoiditis, empyema, meningitis, intracranial abscess or Lemierre's syndrome. A cluster-randomised trial of an antimicrobial stewardship intervention for respiratory prescribing,[10] as well as an interrupted time series analysis found no clear evidence that antimicrobial stewardship policies might be associated with increased bacterial infections overall.[11] However, Gharbi *et al*[12] found that apparent non-use of antibiotics for urinary infections might be associated with higher risk of sepsis.

It is important to extend these investigations to include antibiotic prescribing for all indications because the reasons for antibiotic prescribing may not always be well documented, with up to half of antibiotic prescriptions in UK primary care not associated with any record of specific diagnostic medical codes.[3 7] When analyses are restricted to antibiotic prescriptions for clearly recorded indications, the true extent of antibiotic prescribing may be underestimated. It is also important to assess repeat antibiotic prescriptions which may be given for prevention of recurrent infections or treatment of serious or chronic infections.[3] The present study aimed to test the hypothesis that greater use of antibiotics for all indications might be associated with lower risk of serious bacterial infection. We also investigated whether patterns of medical coding were associated with the apparent occurrence of serious bacterial infection.

## METHODS

### Data source

We carried out a population-based cohort study in the UK Clinical Practice Research Datalink (CPRD) employing data for 2002–2017. The CPRD is one of the world's largest databases of primary care electronic health records, with participation of about 7% of UK family practices and with ongoing collection of anonymised data from 1990.[13] The high quality of CPRD data has been confirmed in many studies.[14] In order to estimate family practice-level prescribing metrics, we analysed a sample of CPRD data. This was because it was not feasible to analyse all antibiotic prescription for the whole of CPRD because the resulting dataset would have been too large for analysis. However, we ascertained serious bacterial infection events from the entire population of CPRD because these are generally rare events. The protocol for the study has been published.

### Selection of sample for antibiotic prescribing analysis

In order to analyse antibiotic prescribing, a sample was drawn from the CPRD denominator file for the October 2018 release of CPRD. A random sample of registered patients was drawn, stratifying by year between 2002 and 2017 and by family practice. In each year of study, a sample of 10 participants was taken for each gender and age group using 5-year age groups up to a maximum of 104 years. Each sampled participant contributed data in multiple years of follow-up. There was a total sample of 671830 individual participants, registered at a total of 706 family practices, who contributed person time between 2002 and 2017. The sampling design enabled estimation of all age-specific rates with similar precision, while age-standardisation provided weightings across age groups.

### Main measures for antibiotic prescribing

For each participant in the antibiotic prescribing sample, we calculated the person-time at risk between the start and end of the patient's record. Person time was grouped by gender, age group and comorbidity. Age groups were from 0 to 4, 5 to 9 and 10 to 14 and then 10 years age groups up to 85 years and over. Comorbidity was evaluated as either present or absent in each person-year using the 'seasonal influenza at risk codes' which are used to identify individuals at higher risk of infection who may benefit from influenza vaccination,[15] as reported previously.[10] Seasonal influenza at risk Read codes include medical diagnostic codes for overweight and obesity, coronary heart disease, chronic kidney disease, chronic liver disease, chronic neurological disease, chronic respiratory disease, diabetes mellitus and disorders of the immune system and drug product codes for asthma therapy, corticosteroid drugs and immunosuppressive drugs. Conditions were coded as present if they were ever diagnosed up to the end of the study year. Collectively, these provide a summary measure of potential susceptibility to infection complications.

Antibiotic prescriptions were evaluated using product codes for antibiotics listed in section 5.1 of the British National Formulary, excluding methenamine and drugs for tuberculosis, and leprosy. Different antibiotic classes and antibiotic doses were not considered further in this analysis. Multiple antibiotic prescription records on the same day were considered as a single antibiotic prescription. Medical codes recorded on the same date as the antibiotic prescription were used to classify the indication for prescription using categories of 'respiratory', 'genitourinary', 'skin' and 'other specific' indications. All other codes were classified as 'non-specific' codes.[3] A prescriptions was classified as 'acute' if it was the first prescription in a sequence or 'repeat' prescription otherwise, as reported previously.[3] Antibiotic prescriptions that were not associated with medical codes and were not repeat prescriptions were classified as 'no codes recorded'.

### Serious bacterial infections

Incident cases of serious bacterial infection were evaluated in the January 2019 release of CPRD for the years

**Table 1** Groups of serious bacterial infections including numbers of medical codes and five most frequently recorded conditions

| Group | No of codes | No of first events | Five most frequent conditions (no of first events 2002–2017) |
|---|---|---|---|
| CNS infection | 30 | 576 | Epidural abscess (117), cerebral abscess (112), brain abscess (79), intraspinal abscess (49), drainage of abscess of subdural space (44) |
| CVS infection | 24 | 1697 | Acute and subacute endocarditis (594), bacterial endocarditis (276), subacute bacterial endocarditis (270), acute endocarditis NOS (166), acute bacterial endocarditis (114) |
| Kidney infection | 22 | 30 827 | Acute pyelonephritis (19 284), pyelonephritis unspecified (7115), infections of kidney (1670), acute pyelitis (1008), pyelitis unspecified (745) |
| Lung abscess/empyema | 24 | 2932 | Empyema (2314), abscess of lung (149), abscess of lung and mediastinum (139), thorax abscess NOS (68), pleural empyema (56) |
| Mastoiditis | 10 | 1970 | Mastoiditis and related conditions (1293), mastoiditis NOS (487), acute mastoiditis (146), acute mastoiditis NOS (31), abscess of mastoid[27] |
| Osteomyelitis | 65 | 4921 | Acute osteomyelitis (3297), unspecified osteomyelitis (678), unspecified osteomyelitis of unspecified site (284), osteomyelitis jaw (78), unspecified osteomyelitis NOS (75) |
| Peritonsillar abscess | 6 | 11 338 | Quinsy (8611), peritonsillar abscess—quinsy (1748), O/E quinsy present (654), drainage of peritonsillar abscess (232), drainage of quinsy (226) |
| Resistant infections and *Clostridium difficile* | 31 | 42 185 | *C. difficile* toxin detection (20 175), methicillin-resistant *Staphylococcus aureus* positive (9914), *C. difficile* infection (6397), methicillin-resistant *S. aureus* (4303), methicillin-resistant *S. aureus* carrier (1017) |
| Sepsis | 100 | 39 059 | Sepsis (23 149), septicaemia (6204), urosepsis (4646), biliary sepsis (1233), *C. infection* (576) |
| Septic arthritis | 41 | 4254 | Septic arthritis (3649), pyogenic arthritis (184), arthropathy associated with infections (172), knee pyogenic arthritis (52), staphylococcal arthritis and polyarthritis (39) |

Figures are frequencies.
CNS, central nervous system; CVS, cardiovascular system; NOS, not otherwise specified.

2002–2017 with the CPRD denominator providing the person time at risk. CPRD records include details of consultations by general practice staff, as well as coded records of referrals to hospital or discharge letters from hospitals. The mean duration of follow-up was 6.9 years. Serious bacterial infections were selected for study from review of the International Classification of Diseases 10th revision,[16] the Read code classification[17] and through discussion with the research team. The final list of conditions is summarised in table 1 and included: bacterial infections of the central nervous system; bacterial infections of the cardiovascular system; kidney infections; lung abscess and empyema; mastoiditis; osteomyelitis; peritonsillar abscess; resistant infections and *Clostridium difficile*; sepsis and septic arthritis. Incident events were first records for each type of serious bacterial infection in a patient more than 12 months after the start of the patient record. However, a single patient might have first episodes of more than one type of bacterial infection. Possible recurrent events in the same patient were not evaluated further because, in electronic health records, it may not be possible to distinguish new occurrences from reference to ongoing or previous problems.

### Statistical analysis

The analysis was in two stages. First, we estimated family practice-specific estimates for antibiotic prescribing; second, we evaluated whether these estimates were associated with the risk of serious bacterial infection. In the first stage of the analysis, we analysed antibiotic prescribing in primary care between 2002 and 2017 (online supplementary table 1: model 1). A hierarchical Poisson model was fitted using the 'hglm' package in the R programme,[18] with counts of antibiotic prescriptions as the outcome and the log of person time as the offset. Estimates were adjusted for the fixed effects of gender, age group, fifth of deprivation at family practice level, comorbidity and region in the UK. Calendar year was included as a continuous predictor together with quadratic and cubic terms to allow for nonlinear trends. Random intercepts were estimated for each family practice and each estimate represented the adjusted log relative rate for antibiotic prescribing at that practice compared with the overall mean. The proportion of antibiotic prescriptions that were associated with specific medical codes was analysed in a similar framework with coded prescriptions as the outcome and the log of antibiotic prescriptions as the offset.

In the second stage of analysis, serious bacterial infections were analysed as the outcome (online supplementary table 1: model 2). The antibiotic prescribing level for each family practice was included as a predictor using the family practice-specific estimates from model 1. These estimates initially had a mean of 0 and SD of 0.19, consistent with an adjusted relative rate of antibiotic prescribing of 1.21 for a family practice with prescribing 1 SD above the mean. Estimates were, therefore, standardised to give the change in serious bacterial infection for a 20% relative increase in antibiotic prescribing rate at a practice, because this represents a change of approximately 1 SD. A 20% change generally represents a substantial change in antibiotic prescribing. We also estimated the change in serious bacterial infection for a 20% relative increase in proportion of antibiotic prescriptions with specific medical codes recorded at a family practice. Models were adjusted for age group, gender, region, deprivation fifth, calendar year (including quadratic and cubic terms for the latter), with log of person-time as offset. The results were visualised using forest plots.[19]

### Patient and public involvement

The protocol and results of the study were discussed at meetings with patients. Patients commented on the recent declining trend in antibiotic prescribing. They noted that avoiding antibiotics requires trade-offs between the limited benefits from antibiotic treatment, the side effects of antibiotic use, and the potential from longer-term problems from the increase in antimicrobial resistance. Patients considered that risks of serious bacterial infections were generally low at the present time. There is a need to communicate these results to patients and prescribers so that both groups can be aware of the wider contextual issue of antimicrobial resistance to inform antibiotic prescribing decisions.

### RESULTS

There were 706 family practices included in the analysis, with 10.1 million registered patients and 69.3 million patient-years of follow-up. In the subsample analysed for antibiotic prescribing, there were 706 family practices with 6 541 195 person-years of follow-up (online supplementary figure 1 and online supplementary table 2). There were a total of 4 371 715 antibiotic prescriptions between 2002 and 2017. This included 2 368 551 (54%) with coded indications including 1 531 645 (35%) associated with respiratory infections, 369 389 (8%) with genitourinary infections, 414 680 (10%) with skin infections and 52 837 (1%) with other specific indications. There were 2 003 164 (46%) of antibiotic prescriptions without specific coded indications consisting of 479 421 (11%) repeat prescriptions, 1 154 789 (26%) with non-specific medical codes recorded and 368 954 (8%) with no medical codes recorded.

Online supplementary figure 2 shows changes over time in age-standardised antibiotic prescribing rates per

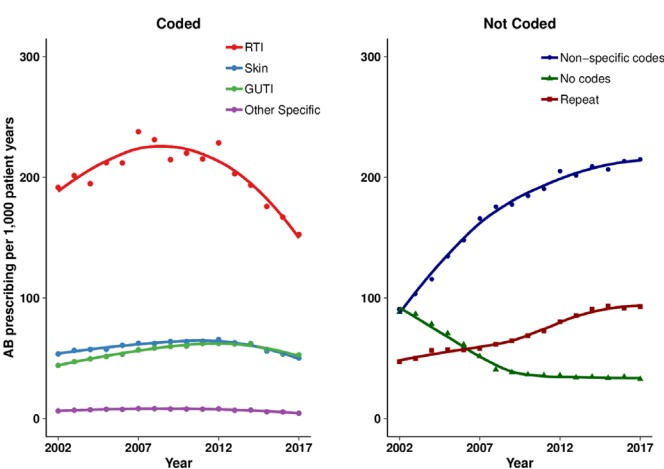

**Figure 1** Age-standardised and sex-standardised antibiotic prescribing rates per 1000 patient-years for coded and not coded indications from 2002 to 2017. AB, antibiotic; GUTI, genito-urinary tract infection; RTI, respiratory tract infection.

1000 patient-years for coded and not coded indications. During the initial period of the study from 2002 to 2012, the age-standardised total antibiotic prescribing rate per 1000 patient-years increased from 2002 (male 423; female 621) to 2012 (male 530; female 842) before declining to 2017 (male 449; female 753). The recent decrease in total antibiotic prescribing was accompanied by a decline in antibiotic prescribing for coded indications, but antibiotic prescriptions that were not associated with specific coded indications continued to increase. There was evidence of a decline in antibiotic prescribing for respiratory illness from 2008 onwards (figure 1) and after 2012 there was evidence of decreasing prescribing for genitourinary and skin infections, as well as other specific indications. Throughout the period from 2002 to 2017, antibiotic prescriptions associated with non-specific codes increased as did repeat prescriptions. Antibiotic prescriptions that were not associated with medical codes declined initially but then remained constant (figure 1).

Table 2 summarises variation in antibiotic prescribing metrics between family practices in the sample. The 95% range for family practice-specific antibiotic prescribing rates was from 430 to 1038 antibiotic prescriptions per 1000 person-years, with a median of 648 antibiotic prescriptions per 1000 patient-years. The 95% range for the proportion of repeat prescriptions was from 3% to 24%. The 95% range for the proportion of antibiotic prescriptions with specific coded indications recorded ranged from 10% to 75%.

There were 139 759 first episodes of serious bacterial infections (online supplementary table 3). Figure 2 shows trends in the age-standardised incidence of serious bacterial infections from 2002 to 2017. The total incidence of serious bacterial infections increased during the period. This increase was largely accounted for by increases in sepsis, antibiotic resistant and *C. difficile* infections, kidney infections and osteomyelitis. The remaining conditions showed either stable incidence or slight declines. Online supplementary

**Table 2** Variation in antibiotic prescribing between family practices

| Measure | Centiles of family practices | | | | |
| --- | --- | --- | --- | --- | --- |
| | 2.5th | 25th | Median | 75th | 97.5th |
| AB prescribing rate per 1000 patient-years | 430 | 563 | 648 | 748 | 1038 |
| Acute prescriptions (% of all antibiotic prescriptions) | 76 | 86 | 90 | 93 | 97 |
| Repeat prescriptions (% of all antibiotic prescriptions) | 3 | 7 | 10 | 14 | 24 |
| Coded indication (% of all antibiotic prescriptions) | 10 | 48 | 58 | 65 | 75 |
| Respiratory (% of all antibiotic prescriptions) | 6 | 31 | 36 | 42 | 52 |
| Genitourinary (% of all antibiotic prescriptions) | 1 | 7 | 8 | 11 | 16 |
| Skin (% of all antibiotic prescriptions) | 2 | 8 | 10 | 12 | 16 |
| Other specific (% of all antibiotic prescriptions) | 0 | 1 | 1 | 2 | 3 |
| Non-coded indications (% of all antibiotic prescriptions) | 24 | 35 | 42 | 51 | 90 |
| No codes recorded (% of all antibiotic prescriptions) | 1 | 3 | 6 | 11 | 28 |
| Non-specific codes recorded (% of all antibiotic prescriptions) | 12 | 19 | 24 | 29 | 59 |

Column per cents are not expected to sum to 100 as different family practices may be represented for the same centile in different rows.
Figures represent the centiles of the distribution of family practice-specific values.
AB, antibiotic.

table 4 presents age-standardised and sex-standardised incidence rates per 1000 patient-years for serious bacterial infections for the highest and lowest fourths of antibiotic prescribing. There was no evidence that serious bacterial infections might be more frequent at family practices in the lowest fourth of antibiotic prescribing. In general, age-standardised and sex-standardised incidence rates tended to be highest at family practices that were higher prescribers of antibiotics. Online supplementary table 4 also compares the incidence of serious bacterial infection for the lowest and highest fourths of medical coding. In the lowest quartile of practices a median of 38% antibiotic prescriptions were coded, compared with 70% for practices in the highest quartile. Family practices in the highest fourth of medical coding had an incidence of serious bacterial infection of 2.39 per 1000 patient-years (95% CI 2.37 to 2.42) compared with 1.94 (1.91 to 1.96) in the lowest fourth of medical coding.

Figure 3 presents a forest plot for the association of each serious bacterial infection with 20% higher total antibiotic prescribing at a family practice. The combined estimate revealed that there was no evidence that higher total antibiotic prescribing was associated with lower incidence of serious bacterial infections (adjusted rate ratio (RR) 1.03, 95% CI 1.00 to 1.06, p=0.074). When the 10 classes of serious bacterial infection were considered individually, there was no evidence that higher antibiotic prescribing might be associated with a lower incidence of infections.

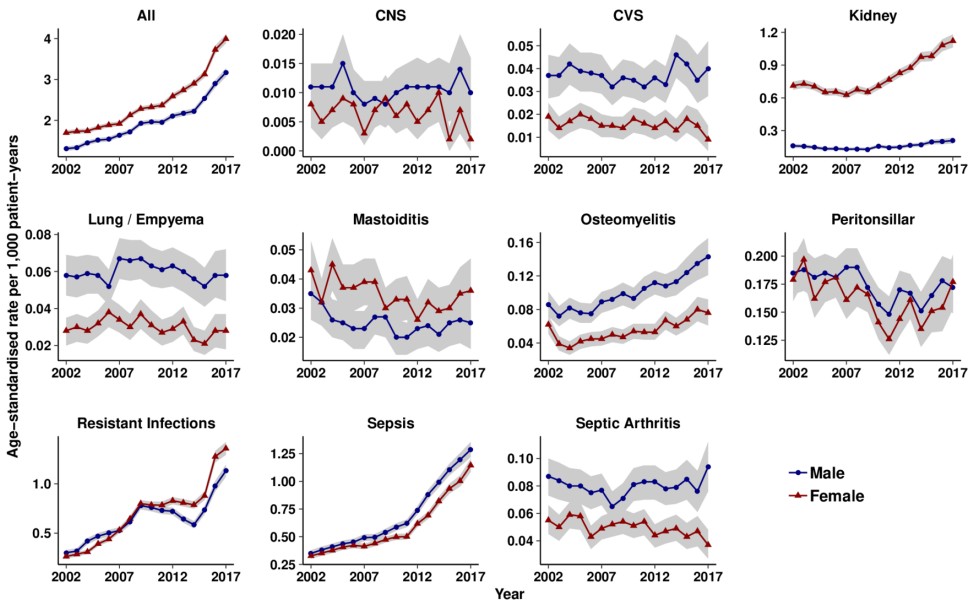

**Figure 2** Age-standardised rates of serious bacterial infections per 1000 patient-years from 2002 to 2017. Red lines, female; blue lines, male; shaded areas, 95% CIs. CNS, central nervous system; CVS, cardiovascular system.

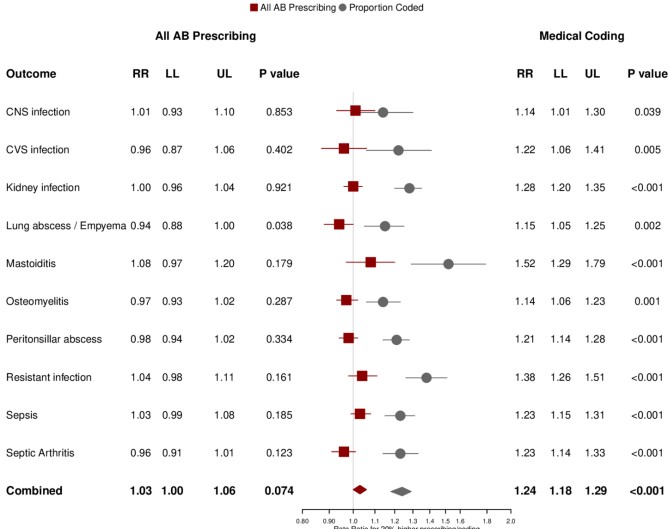

| | All AB Prescribing | | | | | Medical Coding | | | |
|---|---|---|---|---|---|---|---|---|---|
| Outcome | RR | LL | UL | P value | | RR | LL | UL | P value |
| CNS infection | 1.01 | 0.93 | 1.10 | 0.853 | | 1.14 | 1.01 | 1.30 | 0.039 |
| CVS infection | 0.96 | 0.87 | 1.06 | 0.402 | | 1.22 | 1.06 | 1.41 | 0.005 |
| Kidney infection | 1.00 | 0.96 | 1.04 | 0.921 | | 1.28 | 1.20 | 1.35 | <0.001 |
| Lung abscess / Empyema | 0.94 | 0.88 | 1.00 | 0.038 | | 1.15 | 1.05 | 1.25 | 0.002 |
| Mastoiditis | 1.08 | 0.97 | 1.20 | 0.179 | | 1.52 | 1.29 | 1.79 | <0.001 |
| Osteomyelitis | 0.97 | 0.93 | 1.02 | 0.287 | | 1.14 | 1.06 | 1.23 | 0.001 |
| Peritonsillar abscess | 0.98 | 0.94 | 1.02 | 0.334 | | 1.21 | 1.14 | 1.28 | <0.001 |
| Resistant infection | 1.04 | 0.98 | 1.11 | 0.161 | | 1.38 | 1.26 | 1.51 | <0.001 |
| Sepsis | 1.03 | 0.99 | 1.08 | 0.185 | | 1.23 | 1.15 | 1.31 | <0.001 |
| Septic Arthritis | 0.96 | 0.91 | 1.01 | 0.123 | | 1.23 | 1.14 | 1.33 | <0.001 |
| Combined | 1.03 | 1.00 | 1.06 | 0.074 | | 1.24 | 1.18 | 1.29 | <0.001 |

**Figure 3** Forest plot showing the adjusted rate ratio for each type of serious bacterial infection for 20% higher total antibiotic prescribing (red) or 20% higher proportion of antibiotic prescriptions with specific coded indications recorded (grey). Estimates were adjusted for each variable shown and gender, age group, comorbidity, deprivation fifth, region and year (including quadratic and cubic terms). CNS, central nervous system; CVS, cardiovascular system; RR, rate ratio; LL, lower limit 95% confidence interval; UL, upper limit 95% confidence interval.

However, there was weak evidence of that lung abscess and empyema (RR 0.94, 0.88 to 1.00, p=0.038) might be lower at higher prescribing family practices. There was strong evidence that the recorded incidence of serious bacterial infections was associated with the coding of specific indications for a antibiotic prescriptions (adjusted RR for a 20% increase in coding proportion 1.24, 1.18 to 1.29, p<0.001). This association held for each of the 10 classes of serious bacterial infections considered individually.

We conducted a sensitivity analysis by excluding repeat prescriptions that might not have been for acute infection episodes. There was no evidence that higher acute (non-repeat) antibiotic prescribing was associated with serious bacterial infections overall (RR 1.02, 0.99 to 1.05, p=0.227) (online supplementary figure 3). There was evidence that higher acute antibiotic prescribing might be associated with lower incidence of lung abscess and empyema and septic arthritis. Osteomyelitis and peritonsillar abscess were not judged to be associated with acute antibiotic prescribing after controlling the false discovery rate. There was weak evidence that higher repeat antibiotic prescribing might be associated with higher incidence of serious bacterial infections overall (RR 1.01, 1.00 to 1.02, p=0.054) with evidence of this association for kidney infections, osteomyelitis, peritonsillar abscess and septic arthritis considered separately.

## DISCUSSION
### Principal findings
This study found that antibiotic prescribing increased from 2002 to 2012 but declined subsequently with changes

over time being of larger magnitude for women than men. The incidence of serious bacterial infections in men and women rose steadily between 2002 and 2017, particularly for sepsis (men and women), osteomyelitis (mainly in men) and kidney infections (mainly in women). The research aimed to test the hypothesis that family practices with lower utilisation of antibiotics might have greater risk of serious bacterial infections. We evaluated the incidence of serious bacterial infections including 10 groups of infections that affect different systems of the body as well as sepsis (including septicaemia). We did not find evidence that family practices that prescribe antibiotics less frequently might have a higher incidence of serious bacterial infections. We found evidence that each type of serious bacterial infection was recorded more frequently at family practices that record diagnostic codes for a high proportion of antibiotic prescriptions suggesting that variation in the incidence of serious bacterial infection among family practices may be partly an artefact of data recording. Measures are needed to improve the recording of infection episodes in primary care both when antibiotics are prescribed and when they are not. Repeat prescriptions account for a significant proportion of uncoded prescriptions[3] and repeat prescriptions might be indicated for prolonged or serious infections. Certain conditions may be associated with a higher rate of repeat antibiotic prescribing if there is initial treatment failure. For example, surgical intervention may eventually be required for treatment empyema, osteomyelitis or infective endocarditis. We conducted analyses after excluding repeat prescriptions and these analyses raised the possibility that family practices with lower acute (non-repeat) antibiotic prescribing might have higher incidence of lung abscess and empyema and septic arthritis. However, these analyses were not preplanned, should be considered as hypothesis generating and requiring confirmation in future studies. The incidence of these two conditions is less than one per 10 000 patients per year, and a relative rate of 0.9 for a 20% increase in prescribing implies that at most one additional case might arise every 10 years from a 20% reduction in prescribing at a family practice with 10 000 registered patients.

### Strengths and weaknesses of the study
The study drew on data for a large population comprising data for about 7% of the UK general population. In view of sample size constraints, antibiotic utilisation was estimated through analysis of data for a sample of patients, using hierarchical (multilevel) regression models to obtain family practice-specific antibiotic prescribing estimates. This contrasts with our previous study in which age-standardised and sex-standardised rates were calculated from the data for each practice.[9] Use of a regression modelling approach enabled us to make optimal use of the data, as well as adjusting for covariates that are associated with variations in antibiotic prescribing[20] including comorbidity, deprivation, region and calendar year, in addition to age and sex.[21] Consistent with previous studies,[3 7] we observed

that nearly half of antibiotic prescriptions were not associated with specific coded indications. This suggests that total antibiotic prescribing is the most appropriate exposure measure for consideration, because indication-specific antibiotic prescribing may be associated with considerable misclassification. Serious bacterial infections were identified from medical diagnostic codes recorded into primary care electronic health records, which include general practice records of consultations, hospital referrals and discharges. Many studies have shown that these records have a high predictive value for a range of diagnoses,[14] but relying on a single data source can lead to underestimation of the total number of events.[22] CPRD records are linked to hospital episode statistics (HES) but only for a subset of general practices in England, leading to a reduced sample size. Further research incorporating HES data are now underway and will be reported separately. There may be changes over time in the use of diagnostic categories, which might in part account for increasing diagnoses of 'sepsis'. A study of US hospitals' data found that there was a 70.6% increase in sepsis between 2003 and 2012, without any corresponding increase in positive blood cultures.[23] There was also an apparent increase in resistant infections but this might also be due in part to data recording changes and growing awareness of the problem of antimicrobial resistance, as well as true increases in resistant infections. An interrupted time series analysis[11] offers an alternative approach to analysis but this might be susceptible to changes over time in unmeasured confounders such as code selection. The results of our study draw attention to the problem of poor coding quality in the context of infection management in primary care. Evidence from other studies suggests that missing values are typically missing not at random and the act of data recording may introduce a form of confounding by indication that may bias results.[24] In order to allow for this, we explicitly evaluated the extent to which differences in data recording between practices might account for variations in the incidence of serious bacterial infections. It is likely that misclassification of exposure and outcome variables, from incomplete data recording, might lead to underestimation of associations, though the direction of bias cannot always be anticipated.[25] We adjusted for a summary measure of comorbidity. Our analyses do not exclude the possibility that there may be vulnerable subgroups of patients, such as those with immunosuppression, who may be at increased risk if antibiotics are withheld.

### Comparison with other studies

The trends in total antibiotic utilisation reported here are consistent with national trends based on aggregate data.[2] Neilly et al[26] found that increasing prescription volumes in the period up to 2013 could be accounted for by increasing dose and duration of prescriptions but we found evidence of increased antibiotic prescribing based on numbers of prescriptions alone. Consistent with our findings, Balinskaite et al[11] reported increasing rates of infection in English primary care and hospital admissions data from 2010 to 2017. Their time series analysis suggested that antimicrobial stewardship intervention in 2015 had no impact on bacterial infections overall but there was some evidence for increasing hospital admissions for quinsy, decreasing hospital admissions for pyelonephritis and decreasing general practitioner consultation rates for empyema. In a previous study, we found that peritonsillar abscess and pneumonia might be more frequent when family practices prescribe antibiotics less frequently for respiratory tract infections.[9] We did not include pneumonia in this study because we found that syndromes of 'chest infection' and 'pneumonia' may be difficult to distinguish in primary care records with evidence of code shifting between the two categories.[27] In the present study, the incidence of peritonsillar abscess was not associated with total antibiotic prescribing. Randomised trials suggest that antibiotics protect against peritonsillar abscess[28] so it is plausible that this condition might be associated with respiratory antibiotic prescribing but not total antibiotic prescribing.

### MAIN CONCLUSIONS

Family practices that reduce the amount of antibiotics prescribed do not risk any increase in serious bacterial infections overall. This finding does not exclude the possibility that serious bacterial infection may be associated with antibiotic prescribing patterns at individual patient level. Consequently, reducing antibiotic utilisation in primary care will require a detailed understanding of when antibiotics prescriptions are required and when they are not and increasing the quality of data recording with respect to antibiotic use should be a high priority. This study focused on population-level associations at the level of family practice. Future research should evaluate the associations at the level of the individual patient and the individual family practice consultation. This might provide primary care professionals and patients with objective evidence concerning levels risk that can inform decisions to prescribe or not prescribe antibiotics.

**Collaborators** The SafeABStudy Group also includes Dr Caroline Burgess, Dr Vasa Curcin and Dr James Shearer.

**Contributors** MCG wrote the study protocol with advice from CB, RF, MA, PL, MM and ADH. XS developed and piloted code sets and analyses for antibiotic prescribing; RF, PL, MM, ADH and MA reviewed case definitions. JC programmed analyses and JRW advised. MCG completed data analyses and drafted the paper with advice from CB, RF, PL, MM, ADH and MA. OB coordinated PPI input. All authors reviewed and contributed to the final draft. MCG is guarantor.

**Funding** The study is funded by the National Institute for Health Research (NIHR) Health Services and Delivery Programme (16/116/46). MCG was supported by the NIHR Biomedical Research Centre at Guy's and St Thomas' Hospitals.

**Disclaimer** The views expressed are those of the authors and not necessarily those of the NHS, the NIHR, or the Department of Health. The funder of the study had no role in study design, data collection, data analysis, data interpretation, or writing of the report. The authors had full access to all the data in the study and all authors shared final responsibility for the decision to submit for publication.

**Competing interests** None declared.

**Patient consent for publication** Not required.

**Ethics approval** The protocol was approved by the CPRD Independent Scientific Advisory Committee (ISAC protocol 18–041R).

**Provenance and peer review** Not commissioned; externally peer reviewed.

**Data availability statement** Data are available on reasonable request. Requests for access to data from the study should be addressed to martin.gulliford@kcl.ac.uk. All proposals requesting data access will need to specify planned uses with approval of the study team and CPRD before data release.

**ORCID iDs**
Martin C Gulliford http://orcid.org/0000-0003-1898-9075
Xiaohui Sun http://orcid.org/0000-0003-1576-4903
Catey Bunce http://orcid.org/0000-0002-0935-3713
Mark Ashworth http://orcid.org/0000-0001-6514-9904

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
