## [Reviewer comments · BMJ Open]

ARTICLE DETAILS

TITLE (PROVISIONAL)	SERIOUS BACTERIAL INFECTIONS AND ANTIBIOTIC PRESCRIBING IN PRIMARY CARE. COHORT STUDY USING ELECTRONIC HEALTH RECORDS IN THE UK
AUTHORS	Gulliford, Martin; Sun, Xiaohui; Charlton, Judith; Winter, Joanne; Bunce, Catey; Boiko, Olga; Fox, Robin; Little, Paul; Moore, Michael; Hay, Alastair; Ashworth, Mark

VERSION 1 – REVIEW

REVIEWER	Michael Durkin Washington University School of Medicine, St. Louis, MO, USA
REVIEW RETURNED	21-Jan-2020

GENERAL COMMENTS	This study aimed to determine if serious bacterial infections were more prevalent in family practice with lower antibiotic prescribing rates. They used the a UK family practice database from 2002 to 2017. The question is appropriate. The data set is robust, and the statistical analyses were robust and included multilevel modeling. The authors found no evidence that serious bacterial infections were more common in family practice settings with lower antibiotic prescribing rates. This is an important finding and should reassure public health policy makers that judicious antibiotic use does not place a patient at higher risk of infection-related adverse events. I found the paper to be well-written. I have only a few comments that require clarification. Comments: 1) I found the figure showing a dramatic increase in resistant infections fascinating. Please consider adding more detail about antibiotic resistance and the importance of outpatient antibiotic stewardship in the discussion section.2) The codes for serious infections would usually be associated with a hospitalization. As the dataset is focused in the outpatient primary care setting, can you clarify if or how hospitalization or emergency department icd-10 codes were captured.3) We would expect certain conditions, which have a higher treatment failure rate, to have a higher rate of repeat antibiotic prescribing. Much of the treatment failure rate is dependent on surgical source control. For example, empyema may need decortication; osteomyelitis may need I&D, and CVS infection may need valve replacement. These issues should be described in more detail in the discussion section.
--

	4) Please describe how far you looked back for ICD-10 diagnosis codes associated with pre-existing comorbidities: such as overweight, obesity, coronary artery disease, etc. I suspect it was 1 year, but the language is unclear.
--	--

REVIEWER	Harry Ahmed, GP and Researcher & Holly Peters, GP Academic Fellow Cardiff University, Wales, UK
REVIEW RETURNED	23-Jan-2020

GENERAL COMMENTS	Thank you for asking me to review this paper. The authors analysed CPRD data to (1) estimate antibiotic prescribing rates over time from a sample of patients registered with CPRD practices, and (2) to assess whether practices with lower antibiotic prescribing rates had a greater incidence of serious bacterial infection. This is a really neat paper, especially the additional analysis that looks at the incidence of serious infection by how well practices code the indication for antibiotic prescriptions. Unsurprisingly, practices that code prescriptions better also code serious infections better and thus show a higher incidence of these infections. Another strength of this paper is the clear and balanced discussion of the strengths and weaknesses of these data and this study that include the issues around coding/misclassification and change/overuse of codes as per those for sepsis. Some minor suggestions for clarifications that I think might help BMJ Open readers: METHODS, Data Source:  1. Suggest switch the sentences about family practice level prescribing metrics and serious bacterial infection events around because you then go on to talk about the sample for the antibiotic prescribing analysis first, and the serious bacterial infections second. 2. From what you have written, I don't think that someone unfamiliar with CPRD would understand how and, more importantly, why the samples for the two analyses were different. Can you make these sections a little clearer? METHODS, Main measures for antibiotic prescribing: Not clear why you used "seasonal flu at risk codes" as a proxy for the presence of comorbidity rather than diagnostic codes for the specific comorbidities you've listed. Can you clarify/justify please? How reliable are the seasonal flu at risk codes? For example, what proportion of people with diabetes have a seasonal flu at risk code? METHODS, Serious bacterial infections: "possible recurrent events in the same patient were not evaluated further..." but please clarify what you did when a patient had more than one different serious infection, e.g., a record for sepsis and then a subsequent one 10 months later for septic arthritis. METHODS, Statistical analysis: Please justify the statement about a 20% change in antibiotic prescribing rate being clinically important – I (and some of my GP Colleagues) would argue that a smaller change (e.g., 10%) could be clinically important in the current climate. RESULTS, paragraph 4: Please clarify what the lowest and highest fourths of medical coding actually mean. MAIN CONCLUSIONS: Individual level associations will be challenging due to indication bias and unmeasured confounding – what will this add to what we already know?
--

VERSION 1 – AUTHOR RESPONSE

Reviewer: 1 Reviewer Name: Michael Durkin

I found the paper to be well-written. I have only a few comments that require clarification.
Thank you for this feedback.

1) *I found the figure showing a dramatic increase in resistant infections fascinating. Please consider adding more detail about antibiotic resistance and the importance of outpatient antibiotic stewardship in the discussion section.*

Thank you for this comment, we now observe (page 15): 'There was also an apparent increase in resistant infections but this might also be due in part to data recording changes, and growing awareness of the problem of antimicrobial resistance, as well as true increases in resistant infections.'

2) *The codes for serious infections would usually be associated with a hospitalization. As the dataset is focused in the outpatient primary care setting, can you clarify if or how hospitalization or emergency department icd-10 codes were captured.*

Thank you, we now explain (page 8) 'CPRD records include details of consultations by general practice staff, as well as coded records of referrals to hospital or discharge letters from hospitals.'

We also comment (page 14) 'Serious bacterial infections were identified from medical diagnostic codes recorded into primary care electronic health records, which include general practice records of consultations, hospital referrals and discharges. Many studies have shown that these records have a high predictive value for a range of diagnoses, (14) but relying on a single data source can lead to under-estimation of the total number of events.(22) CPRD records are linked to hospital episode statistics (HES) but only for a subset of general practices in England, leading to a reduced sample size. Further research incorporating HES data is now underway and will be reported separately.'

3) *We would expect certain conditions, which have a higher treatment failure rate, to have a higher rate of repeat antibiotic prescribing. Much of the treatment failure rate is dependent on surgical source control. For example, empyema may need decortication; osteomyelitis may need I&D, and CVS infection may need valve replacement. These issues should be described in more detail in the discussion section.*

Thank you for this comment, which we now include (page 13) 'Certain conditions may be associated with a higher rate of repeat antibiotic prescribing if there is initial treatment failure. For example, surgical intervention may eventually be required for treatment empyema, osteomyelitis or infective endocarditis.'

4) *Please describe how far you looked back for ICD-10 diagnosis codes associated with pre-existing comorbidities: such as overweight, obesity, coronary artery disease, etc. I suspect it was 1 year, but the language is unclear.*

Thank you, we now explain 'Conditions were coded as present if they were ever diagnosed up to the end of the study year.'

Reviewer: 2 Reviewer Name: Harry Ahmed

This is a really neat paper, especially the additional analysis that looks at the incidence of serious infection by how well practices code the indication for antibiotic prescriptions.

Thank you for this feedback.

METHODS, Data Source:

Suggest switch the sentences about family practice level prescribing metrics and serious bacterial infection events around because you then go on to talk about the sample for the antibiotic prescribing analysis first, and the serious bacterial infections second.

Thank you, we have adopted this suggestion.

From what you have written, I don't think that someone unfamiliar with CPRD would understand how and, more importantly, why the samples for the two analyses were different. Can you make these sections a little clearer?

Thank you, we now explain (p6) 'In order to estimate family practice-level prescribing metrics, we analysed a sample of CPRD data. This was because it was not feasible to analyse all antibiotic prescription for the whole of CPRD because the resulting dataset would have been too large for analysis. However, we ascertained serious bacterial infection events from the entire population of CPRD because these are generally rare events.'

METHODS, Main measures for antibiotic prescribing:

Not clear why you used "seasonal flu at risk codes" as a proxy for the presence of comorbidity rather than diagnostic codes for the specific comorbidities you've listed. Can you clarify/justify please? How reliable are the seasonal flu at risk codes? For example, what proportion of people with diabetes have a seasonal flu at risk code?

Thank you, we now explain more clearly (p7) 'Comorbidity was evaluated as either present or absent in each person-year using the 'seasonal flu at risk codes' which are used to identify individuals at higher risk of infection who may benefit from influenza vaccination,(15) as reported previously.(10) Seasonal flu at risk Read codes include medical diagnostic codes for overweight and obesity, coronary heart disease, chronic kidney disease, chronic liver disease, chronic neurological disease, chronic respiratory disease, diabetes mellitus and disorders of the immune system and drug product codes for asthma therapy, corticosteroid drugs and immunosuppressive drugs. Collectively, these provide a summary measure of potential susceptibility to infection complications.' Thus, it follows that all patients coded with diabetes will be analysed as having comorbidity by this definition.

METHODS, Serious bacterial infections:

"possible recurrent events in the same patient were not evaluated further..." but please clarify what you did when a patient had more than one different serious infection, e.g., a record for sepsis and then a subsequent one 10 months later for septic arthritis.

Thank you, we now clarify (p8): 'However, a single patient might have first episodes of more than one type of bacterial infection.'

METHODS, Statistical analysis:

Please justify the statement about a 20% change in antibiotic prescribing rate being clinically important – I (and some of my GP Colleagues) would argue that a smaller change (e.g., 10%) could be clinically important in the current climate.

Thank you, we have modified the wording to acknowledge this concern. This section now reads (p9) 'Estimates were therefore standardised to give the change in serious bacterial infection for a 20% relative increase in antibiotic prescribing rate at a practice, because this represents a change of approximately one standard deviation. A 20% change generally represents a substantial change in antibiotic prescribing.'

RESULTS, paragraph 4:

Please clarify what the lowest and highest fourths of medical coding actually mean.

Thank you for this comment. We now explain (page 11) 'In the lowest quartile of practices a median of 38% antibiotic prescriptions were coded, compared with 70% for practices in the highest quartile.'

MAIN CONCLUSIONS:

Individual level associations will be challenging due to indication bias and unmeasured confounding – what will this add to what we already know?

Thank you, we now provide clarification by saying (p17) 'Future research should evaluate the associations at the level of the individual patient and the individual family practice consultation. This might provide primary care professionals and patients with objective evidence concerning levels risk that can inform decisions to prescribe or not prescribe antibiotics.'

VERSION 2 – REVIEW

REVIEWER	Michael Durkin Washington University in St. Louis, USA
-----------------	---

REVIEW RETURNED	31-Jan-2020
GENERAL COMMENTS	Accept.
REVIEWER	Harry Ahmed Cardiff University UK
REVIEW RETURNED	02-Feb-2020
GENERAL COMMENTS	Thank you for asking me to review the revised version of this manuscript. The authors have fully addressed the issues from the first review and I have no further comments. I look forward to seeing this paper in the BMJ Open soon.